# Mitochondrial Porin Is Involved in Development, Virulence, and Autophagy in *Fusarium graminearum*

**DOI:** 10.3390/jof8090936

**Published:** 2022-09-04

**Authors:** Xueqin Han, Qingyi Li, Xuenan Li, Xiang Lv, Li Zhang, Shenshen Zou, Jinfeng Yu, Hansong Dong, Lei Chen, Yuancun Liang

**Affiliations:** Key Laboratory of Agricultural Microbiology, College of Plant Protection, Shandong Agricultural University, Tai’an 271018, China

**Keywords:** *Fusarium graminearum*, porin, development, virulence, autophagy

## Abstract

Mitochondrial porin, the voltage-dependent anion-selective channel (VDAC), is the most abundant protein in the outer membrane, and is critical for the exchange of metabolites and phospholipids in yeast and mammals. However, the functions of porin in phytopathogenic fungi are not known. In this study, we characterized a yeast porin orthologue, Fgporin, in *Fusarium graminearum*. The deletion of *Fgporin* resulted in defects in hyphal growth, conidiation, and perithecia development. The *Fgporin* deletion mutant showed reduced virulence, deoxynivalenol production, and lipid droplet accumulation. In addition, the *Fgporin* deletion mutant exhibited morphological changes and the dysfunction of mitochondria, and also displayed impaired autophagy in the non-nitrogen medium compared to the wild type. Yeast two-hybrid and bimolecular fluorescence complementation assays indicated that Fgporin interacted with FgUps1/2, but not with FgMdm35. Taken together, these results suggest that *Fgporin* is involved in hyphal growth, asexual and sexual reproduction, virulence, and autophagy in *F. graminearum*.

## 1. Introduction

*Fusarium graminearum* is one of the main pathogens causing Fusarium head blight (FHB) on wheat and other cereal crops [1], which leads to yield losses and produces secondary metabolic toxins. Importantly, these mycotoxins pose a serious threat to human and animal health [2]. Deoxynivalenol (DON), one of the trichothecene mycotoxins produced by *F. graminearum*, is considered an important virulence factor during plant infection [3,4]. Ascospores are considered to cause primary infection; FHB possibly occurs during wheat flowering and spreads through the air. A lack of disease-resistant varieties and fungicide resistance frequently leads to FHB epidemics, especially under suitable climatic conditions [5].

Mitochondria are key organelles comprising of the inner membrane (IM) and the outer membrane (OM). Mitochondrial porin, the voltage-dependent anion-selective channel (VDAC), is the most abundant protein in OM [6], and facilitates the exchange of ions and small metabolites such as ATP between mitochondria and cytosol, through its β-barrel structure [7,8]. Many organisms possess more than one porin isoform. In *Saccharomyces cerevisiae*, there are two porin homologs, porin1 and porin2. Porin1 is the most abundant protein in the yeast mitochondrial outer membrane, with up to 16,000 copies per mitochondria [9]. If *porin1* is disrupted, the *porin1* mutant cannot grow on non-fermentative carbon sources such as glycerol, due to an impaired respiratory function [10,11,12]. Porin1 deficiency also reduces autophagic capacity, triggers lysosomal dysfunction, and leads to altered vacuolar and lipid homeostasis [13]. Yeast porin is involved in phospholipid metabolism, and porin1 interacts with the IMS (inter membrane space), Mdm35, which forms the lipid transporter complexes with Ups1/2 to transport phosphatidic acid (PA) and phosphatidylserine (PS) from OM to IM for subsequent cardiolipin (CL) and phosphatidylethanolamine (PE) synthesis [14,15,16]. The filamentous fungus *Neurospora crassa* has a single porin; the deletion of *porin* displays reduced growth and hyphal defects [17]. There are three porins in mammals, mammalian VDAC members are found to play roles in male reproduction, the central nervous system, glucose homeostasis, apoptosis, and carcinogenesis [18,19,20,21].

In this study, one Fgporin in *F. graminearum* was investigated and identified. The results showed that the Δ*porin* mutant was defective in hyphal growth, conidiation, and perithecia formation. More importantly, the Δ*porin* mutant significantly reduced plant infection, mitochondrial function, and autophagy.

## 2. Materials and Methods

### 2.1. Fungal Strains and Culture Conditions

In this study, *Fusarium graminearum* wild type strain PH-1 was used as the parent for transformation experiments to construct gene deletion mutants. Mutant strains were screened using TB_3_ medium supplemented with hygromycin B (Roche, San Francisco, CA, USA) [22]. Strains were cultured on potato dextrose agar (PDA) medium, complete medium (CM), and 1.5% glycerol medium (containing only non-fermented carbon sources; 1.5% glycerol, 0.1 g/L casein tryptone, 0.05 g/L yeast extract, 0.05 g/L casein acid hydrolysate, 0.3 g/L NaNO_3_, 0.025 g/L KCl, 0.25 g/L MgSO_4_·7H_2_O, 0.075 g/L KH_2_PO_4_, and 9 g/L agar) at 25 °C for 3 days. For conidial production, a 4-day-old culture in CMC (carboxymethylcellulose) liquid medium was harvested [22]. The formation of the perithecia was assayed on carrot agar medium, and 7-day-old aerial hyphae were gently pressed down on carrot agar plates after the addition of 2.5% Tween 20 solution [23]. TBI (trichothecene biosynthesis induction) liquid medium was used for DON toxin production [24]. Each experiment was repeated three times.

### 2.2. Construction of Gene Deletion and Complementation Strains

The split-marker approach and PEG-mediated protoplast transformation [25,26] were employed to construct *Fgporin* deletion mutants. Hygromycin-resistant transformants were identified using PCR with the specified primers (Appendix A) and Southern blotting. Southern blot hybridization was prepared using the High Prime DNA Labeling and Detection Starter kit I (Roche Diagnostics, Mannheim, Germany) to confirm single-copy integration. For complementary analysis, a recombinant pFL2 vector containing the full-length *Fgporin* with its promoter sequence was introduced into the protoplasts of the Δ*porin* mutant; after geneticin (BBI Life Sciences Ltd., Beijing, China) screening on PDA plates, geneticin-resistant transformants were verified by PCR as the complemented strains. Similar approaches were used to generate FgUps1/2 and Fgmdm35 mutants.

### 2.3. Plant Infection and DON Production Assays

After incubation in CMC medium for 4 days, conidia were collected and subsequently resuspended in sterile distilled water to a concentration of 2 × 10^5^/mL. Virulence determination was performed on flowering wheat heads and the coleoptiles of wheat cultivars Jimai 22 through point-inoculating conidial suspension. Middle spikelets were inoculated with 10 µL of conidial suspension during the flowering stage, as previously described [27], and the number of infected spikelets was recorded at 14 days post-inoculation (dpi). A 2 µL aliquot of conidial suspension was inoculated on the coleoptile and the infection length was detected at 10 dpi. To determine the production of DON, 200 µL conidial suspension (1 × 10^5^/mL) was added into TBI medium and cultured in the dark for 7 days [24]; the DON content in the culture medium was determined using the Fusarium deoxynivalenol detection kit (Beijing Anyi Century Trading Co., Ltd., Beijing, China).

### 2.4. Quantitative Reverse Transcription-Polymerase Chain Reaction (qRT-PCR) Assays

Total RNA was isolated from hyphae collected from the TBI medium using RNAiso reagent (TaKaRa, Dalian, China). The expression of *TRI5*, *TRI6*, and *TRI10* genes was detected using primer pairs TRI5-F/TRI5-R, TRI6-F/TRI6-R, and TRI10-F/TRI10-R (Appendix A), respectively. The *GAPDH* gene of *F. graminearum* was used as the internal control. The relative expressions of *TRI5*, *TRI6*, and *TRI10* were calculated using the 2^−ΔΔC^^t^ method [28].

### 2.5. Mitochondrial Membrane Potential, ATP, and H_2_O_2_ Analysis

Analysis of mitochondrial function was conducted using a mitochondrial membrane potential (MMP) detection kit, an ATP assay kit, and a H_2_O_2_ assay kit (Biyuntian Biotechnology Research Institute, Shanghai, China) according to the manufacturer’s instructions. Briefly, mycelia stained with JC-1 solution were examined under a fluorescent microplate to determine MMP by the change of the florescence intensity. A total of 0.05 g mycelial samples were collected and the corresponding lysis buffer was added. After lysis, the solution was detected with a fluorescent microplate reader. The quantification of ATP and H_2_O_2_ were determined following the manufacturer’s instructions, respectively. Experiments were repeated three times.

### 2.6. Fluorescence and Transmission Electron Microscopy Assays 

Lipid droplets in conidia and mycelia were stained with 25 µg/mL Nile red solution and then examined via laser scanning confocal microscopy (LSM 880 NLO, Zeiss, Oberkochen, Germany). pFL2-TRI1-GFP and pFL2-TRI4-GFP plasmids were transferred to the protoplasts of PH-1 and Δ*porin* strains, respectively, and the correct transformants were obtained using PCR and fluorescence screening. After 3 days of incubation in TBI liquid medium, the formation of toxisomes in the hyphae was observed under the confocal microscope. For transmission electron microscopy (TEM) assays, the hyphae of PH-1 and the Δ*porin* strain were washed twice with phosphate-buffered saline (PBS) for 10 min, stained with osmium tetroxide for 1 h, and washed twice with PBS and once with 0.1 N acetate buffer for 10 min. The samples were embedded using the Spurr Low Viscosity Embedding kit, and stained with 1% uranyl acetate and 0.4% lead citrate. The samples were visualized using an accelerating voltage of 80 kV on a FEI Tecnai G2 Twin TEM (FEI, Hillsboro, OR, USA).

### 2.7. Autophagy Detection

Transformation of the vector carrying GFP-FgAtg8 into the protoplasts of PH-1 and Δ*porin* strains was performed using PEG-mediated transformation methods. After the incubation of the PH-1/GFP-FgAtg8 and Δ*porin*/GFP-FgAtg8 strains in CM liquid medium at 25 °C for 24 h, the hyphae were transferred to MM-N (minimal medium without nitrogen) liquid medium for 6 h, and stained with 10 μM CMAC (7-amino-4-chloromethylcoumarin) solution [29,30]; autophagy was observed via confocal microscopy fluorescence. For the Western blot assays, total protein extracts were prepared using the Filament Protein Extraction Kit (Shanghai Beibo Biotechnology Co., Ltd., Shanghai, China), isolated via electrophoresis in a 10% SDS-PAGE gel, transferred to a PVDF membrane, and then labeled using GFP-specific monoclonal antibodies (Cell Signaling Technology, Danvers, MA, USA). GAPDH was used as an internal protein control detected using GAPDH-specific monoclonal antibodies (Huaan Biotechnology, Hangzhou, China).

### 2.8. Yeast Two-Hybrid and Bimolecular Fluorescence Complementation (BiFC) Assays

For yeast two-hybrid assays, using the full-length cDNA of *Fgporin*, *FgUps1* (FGSG_10309), *FgUps2* (FGSG_10319), and *FgMdm35* (FGSG_06057) as templates to construct recombinant plasmids, the resulting vectors of pGADT7-Fgporin, pGADT7-FgUps1/2, pGADT7-FgMdm35, and pGBKT7-Fgporin were constructed, respectively. The recombinant plasmids were co-transferred into the yeast strain AH109 [31]; transformants were coated on a SD-Trp-Leu solid medium and then cultured on SD-Trp-Leu-His-Ade solid medium at 28 °C, as described previously [32]. The positive and negative controls were provided in the Matchmaker library construction kit (Clontech, San Jose, CA, USA).

For the BiFC assays, pHZ65 and pHZ68 plasmids with the N and C terminals of the YFP protein were applied, respectively. The Fgporin-YFP^N^, FgUps1-YFP^N^, FgUps2-YFP^N^, FgMdm35-YFP^N^, and Fgporin-YFP^C^ fusion expression vectors were constructed. The recombinant plasmids were co-transferred into PH-1 in pairs. Transformants were generated via geneticin resistance screening and PCR, and YFP signals in the hyphae were examined using laser confocal fluorescence.

## 3. Results

### 3.1. Identification of Fgporin in F. graminearum

Using the amino acid sequences of *Saccharomyces cerevisiae* porin1 (NP_014343.1) and porin2 (NP_012152.3) as templates, the Blastp sequence alignment tool was used to obtain the predicted *F. graminearum* Fgporin (FGSG_09933), which encoded 283 amino acids. Through amino acid sequence alignment, it was found that Fgporin was 100% similar to porin of *F. oxysporum*, 82% similar to porin of *N. crassa*, and 43% and 33% similar to porin1 and porin2 of *S. cerevisiae*, respectively (Figure 1A, Appendix A). Secondary structure analysis showed that Fgporin had one α-helix and 19 β-sheets, and that the β-sheets formed a barrel structure, that was similar to the secondary structure of *S. cerevisiae* porin1 and porin2 (Figure 1B). In order to further determine the homology between Fgporin and other fungal porins, multiple fungal porin proteins were selected, and the results showed that porins of *F. graminearum* and *F. oxysporum* had the closest relationship among nine fungi (Figure 1C).

### 3.2. Generation of Fgporin Deletion Mutants and Complementation Strains

To characterize the functions of Fgporin, *Fgporin* deletion mutants were obtained via homologous recombination (Appendix A). Transformants were screened via hygromycin B resistance medium, and further identified using PCR and Southern blot. Four independent *Fgporin* deletion mutants were generated (Appendix A), and one of the mutants (Δ*porin*) was used for the subsequent experiments. To confirm that the phenotypic alterations of the Δ*porin* mutant were caused by the deletion of *Fgporin*, complementation experiments of Δ*porin* strain were performed, and the resulting transformants were screened via geneticin, and verified using PCR (Appendix A), and one of the transformants was regarded as the complemented strain Δ*porin*-C. 

### 3.3. The Δporin Mutant Is Defective in Mycelial Growth, Conidiation, and Sexual Reproduction

To investigate the biological roles of Fgporin in *F. graminearum*, the growth and reproduction of the Δ*porin* mutant were examined. PH-1, Δ*porin* and Δ*porin*-C strains were cultured on PDA, CM, and 1.5% glycerol plates for 3 days, and the colony diameters were determined. The results showed that the Δ*porin* mutant was defective in colony growth and reduced the ability to utilize non-fermentable carbon sources (Figure 2A). Colony growth was reduced by 40.7%, 46.1%, and 69.1% on PDA, CM, and 1.5% glycerol media, respectively, compared to PH-1 (Figure 2B), indicating that *Fgporin* is involved in the regulation of growth. After 4 days of incubation in CMC liquid medium, the conidiation of the Δ*porin* mutant was decreased by 81.5% compared with PH-1 (Figure 2C). In addition, the Δ*porin* strain was found to produce smaller conidia (Table 1); most abundant conidia of the Δ*porin* strain had only three septa. By contrast, the conidia of PH-1 mostly had five septa (Figure 2D,E). The sexual reproduction was also assayed on carrot agar plates. As shown in Figure 2F, both the PH-1 and Δ*porin*-C strains normally produced a large number of black perithecia, but the Δ*porin* mutant failed to produce perithecia until 20 days post-fertilization, indicating that the *Fgporin* mutant is defective in sexual reproduction. Therefore, these results indicate that *Fgporin* is involved in hyphal growth, conidiation, and sexual reproduction in *F. graminearum*.

### 3.4. Δporin Strain Reduces Virulence and DON Production

To determine the effect of *Fgporin* on virulence, wheat heads and coleoptiles were inoculated with conidial suspension. Wheat heads inoculated with the Δ*porin* strain displayed fewer necrotic spikelets and had a smaller disease index on the wheat heads, while the PH-1 and Δ*porin*-C strains could infect most of the spikelets after 14 days of inoculation (Table 1; Figure 3A). Coleoptiles inoculated with the Δ*porin* strain developed brown lesions only at the inoculation sites, whereas PH-1 and Δ*porin*-C strains caused longer lesions on the coleoptiles (Table 1; Figure 3B). To prove whether the decline in virulence is caused by DON, the DON production was assayed. The results showed that the DON toxin content in the Δ*porin* strain was decreased by 96.6% compared with the wild-type PH-1 (Figure 3C). To further demonstrate the effects of *Fgporin* on DON toxin production, *TRI* expression levels were assayed using qRT-PCR. The results showed that the expression levels of *TRI5*, *TRI6*, and *TRI10* genes in the Δ*porin* strain were significantly decreased compared with the PH-1 strain (Figure 3D). These results indicate that the deletion of *Fgporin* significantly reduces virulence and DON production in *F. graminearum*.

### 3.5. Fgporin Is Involved in the Formation of Toxisomes and Lipid Droplets 

To understand whether Fgporin regulates the formation of toxisomes, fluorescent toxisomes were observed in the Δ*porin* strain. After incubation in TBI liquid medium for 3 days, the toxisome formation in hyphae was observed using a laser confocal microscope. The results showed that PH-1 was able to form a large number of crescent toxisomes in the hyphae, while the toxisomes were almost invisible in the Δ*porin* strain, indicating that *Fgporin* is directly involved in the formation of toxisomes (Figure 4A). To further examine whether *Fgporin* regulates the biosynthesis of lipid droplets (LDs) in *F. graminearum*, the hyphae of the PH-1 and the Δ*porin* strains were stained with Nile red. As shown in Figure 4B, LDs were synthesized in large quantities in PH-1 compared to the Δ*porin* strain. In the TBI medium, only fewer LDs were formed in the Δ*porin* strain, indicating that *Fgporin* regulates LD biosynthesis.

### 3.6. The Δporin Strain Results in Defects in Mitochondrial Morphology and Function

To investigate whether Fgporin is important for mitochondria, the mitochondrial morphology and functions were examined via transmission electron microscopy (TEM) and biochemical experiments. The TEM results showed that the mitochondria in the Δ*porin* strain were malformed and had fewer tubular cristae (Figure 5A). To verify mitochondrial functions in the Δ*porin* strain, mitochondrial membrane potential, endogenous H_2_O_2_, and ATP production were analyzed. The results showed that the Δ*porin* strain cultured in CM and TBI medium displayed an approximate 20% decrease in membrane potential, an approximate 63% decrease in endogenous H_2_O_2_, and an approximate 35% decrease in ATP production compared to PH-1 (Figure 5B–D). Overall, *Fgporin* is essential for normal mitochondrial morphology and functions. 

### 3.7. The Fgporin Deletion Mutant Is Defective in Autophagy 

To determine whether or not the deletion of *Fgporin* affects autophagy, an autophagy marker fusion gene *GFP-FgAtg8* was introduced into PH-1 and the Δ*porin* mutant to monitor autophagic flux. GFP fluorescence in the mycelia was examined using confocal microscopy, and the accumulation of GFP-FgAtg8 proteolysis was achieved through Western blot assays. For fluorescence examination, more green fluorescence dots representing autophagosomes existed in vacuoles in PH-1 than that in the Δ*porin* strain grown in MM-N medium for 6 h (Figure 6A,B). As shown in Figure 6C, few GFP-FgAtg8 degradations occurred in Δ*porin* grown in MM-N medium for 6 h, and the Δ*porin* mutant grown in MM-N medium exhibited a lower GFP:GFP-FgAtg8 ratio compared with PH-1 grown in MM-N medium for 6 h. Taken together, these data suggest that the deletion of the *Fgporin* mutant is defective in autophagy in *F. graminearum*.

### 3.8. Fgporin Interacts with Itself and FgUps1/2, but Not with FgMdm35

To investigate whether Fgporin interacts with itself and FgUps1/2-FgMdm35, yeast two-hybrid and BiFC assays were used. The yeast two-hybrid results showed that colonies could grow on SD-Trp-Leu-His-Ade plates for pGADT7-Fgporin+pGBKT7-Fgporin, pGADT7-FgUps1+pGBKT7-Fgporin, pGADT7-FgUps2+pGBKT7-Fgporin, and positive control strains (Figure 7A). Furthermore, BiFC analysis displayed the same interactions as the yeast two-hybrid results. YFP signals were observed in Fgporin–Fgporin and FgUps1/2–Fgporin interactions. However, the YFP signals were not observed in the *FgMdm35-YFP^N^* and *Fgporin-YFP^C^* transformants (Figure 7B). These results indicate that Fgporin exists in self-interaction, and directly interacts with FgUps1/2, but it fails to interact with FgMdm35. To determine the functions of FgUps1/2 and FgMdm35, similar methods to *Fgporin* deletion were carried out. However, only *FgUps2* deletion mutants were obtained, and the results of the phenotypic experiments showed that *FgUps2* had no effect on the growth and virulence of *F. graminearum* (Figure 7C,D). *FgUps1* and *FgMdm35* were unable to obtain mutants through several transformants, indicating that the deletion of *FgUps1* and *FgMdm35* may be lethal.

## 4. Discussion

Mitochondrial porin is the most abundant protein in the outer membrane and is involved in processes such as metabolite exchange and mitochondrial protein import [6]. Porin has three homologs in mammals and two in yeast, and its function has been progressively identified [10,33]. However, to date, little is known about the roles of the porin homologue in plant pathological fungi. In this study, one Fgporin was identified in *F. graminearum*; our findings demonstrate that Fgporin plays important roles in fungal development, virulence, and autophagy.

The *Fgporin* deletion mutant exhibited reduced growth, which was consistent with porins in *S. cerevisiae* and *N. crassa* [13,34]. The Δ*porin* mutant failed to produce perithecia, indicating that *Fgporin* positively regulates sexual reproduction. The Δ*porin* mutant displayed reduced virulence, which was consistent with the decline in DON content. DON is a main virulence factor in *F. graminearum* [3,4]. DON biosynthesis is required for the expression of *TRI* genes; TRI1 and TRI4 proteins have been reported to localize in toxisomes [5,35,36]. The *Fgporin* deletion mutant obviously reduced the production of toxisomes. Studies have shown that toxisomes are also closely associated with lipid droplet formation [36,37]. Lipid droplets, formed at the endoplasmic reticulum in eukaryotes, are important organelles that are responsible for stores of neutral lipids. In *F. graminearum*, lipid droplet biogenesis plays critical roles in development and virulence [37]. Recently, results have shown showed that the deletion of *porin1* resulted in increased lipid droplet content in yeast cells [38]. In contrast to our results, both in mycelia and conidia, the *Fgporin* deletion mutant had relatively lower intracellular lipid droplet content compared to the wild type PH-1. These data suggest that Fgporin plays a critical role in the formation of lipid droplets and toxisomes.

It is known that normal mitochondrial morphology makes for maintaining its functions. In *Taxoplama gondii* and *Drosophila*, the deletion of *porin* resulted in changes of mitochondrial morphology [39,40]. Consistent with this study, the *Fgporin* deletion strain displayed severe morphological changes of mitochondria in *F. graminearum*. Mitochondria are key regulators of redox balance [41]. The main source of endogenous ROS (reactive oxygen species) is a normal by-product of electron leakage in the mitochondrial respiratory chain [42,43]. It is widely accepted that mitochondrial function mainly includes mitochondrial ATP production, MMP, and ROS production, which are regarded as the markers of mitochondrial function. In the present study, both ATP production and MMP were significantly reduced in the Δ*porin* mutant compared to the wild type PH-1, while H_2_O_2_ content was also significantly reduced compared to the wild type, which was consistent with the previous study in yeast [38,44]. These results suggest that the *Fgporin* deletion mutant exhibits abnormal mitochondrial morphology and functions. 

Autophagy, a primordial and highly conserved intracellular process in eukaryotes, is important for the lysosomal (vacuolar) degradation of proteins and membrane recycling. In phytopathogenic fungi, autophagy is closely related to fungal growth and infection [26,45]. More recently, the *porin1* deletion strain displays reduced autophagy in yeast [13]. A previous study showed that autophagy is necessary for plant colonization in *F. gramineaeum* [45]. In this study, the deletion of *Fgporin* also resulted in impaired autophagy, which is likely to provide more metabolites due to mitochondrial dysfunction. 

In yeast, porin and Ups1/2-Mdm35 form lipid transporter complexes to transport PA and PS from OM to IM for subsequent CL and PE synthesis [14,15,16]. The Ups family is highly conserved in eukaryotes; the import of Ups proteins into the IMS is mediated by interactions with Mdm35. Ups–Mdm35 complexes are required for phospholipid metabolism [14]. Interestingly, only one Fgporin was identified in *F. graminearum*, and it formed a dimer via self-interaction. In yeast, porin directly interacts with Mdm35, which also interacts with Ups1/2. However, in *F. graminearum*, Fgporin failed to interact with FgMdm35, while Fgporin directly interacted with FgUps1/2. 

Taken together, our results suggest that mitochondrial porin is involved in hyphal growth, conidiation, sexual reproduction, virulence, mitochondrial function, and autophagy in *F. graminearum*. These results could provide novel targets to control diseases caused by *F. graminearum*.

## Figures and Tables

**Figure 1 jof-08-00936-f001:**
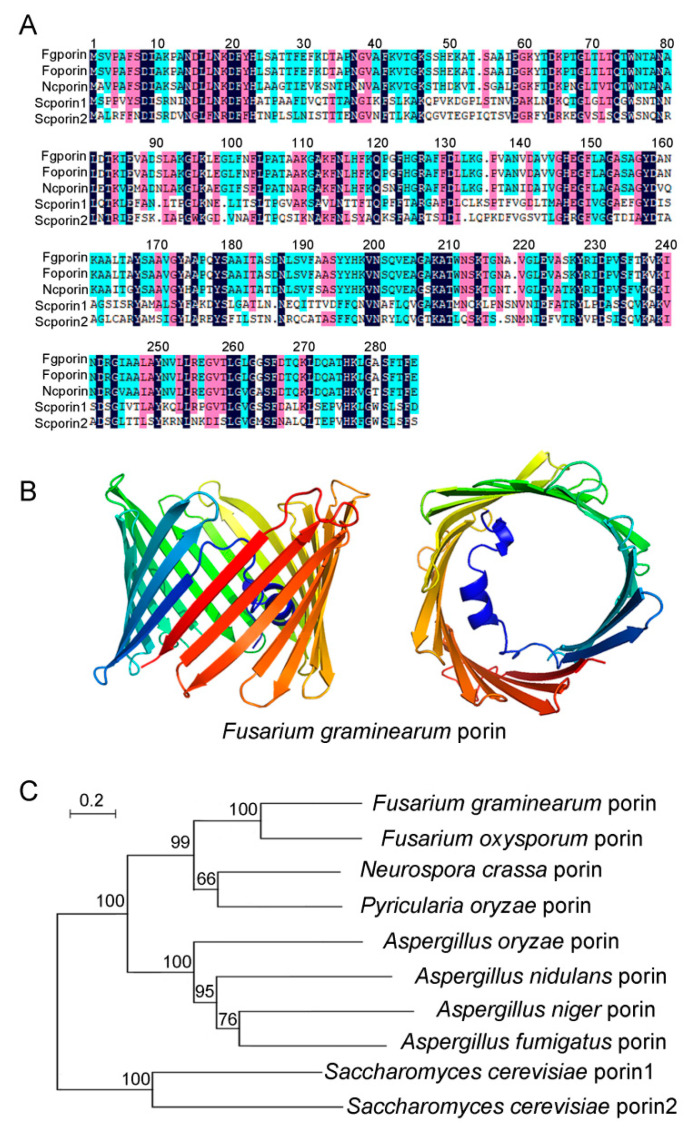
Multiple sequence alignment, secondary structure, and phylogenetic tree analysis of Fgporin. (**A**) Amino acid sequence alignment among *Fusarium graminearum* and other fungal porins. *Fg*, *Fusarium graminearum*; *Fo*, *Fusarium oxysporum*; *Nc*, *Neurospora crassa*; *Sc*, *Saccharomyces cerevisiae*. (**B**) Front and top views of the Fgporin secondary structure. (**C**) Phylogenetic tree analysis of *F. graminearum* Fgporin and other fungal porins. MEGA 7.0 was used to construct a phylogenetic tree based on a conserved sequence alignment pattern.

**Figure 2 jof-08-00936-f002:**
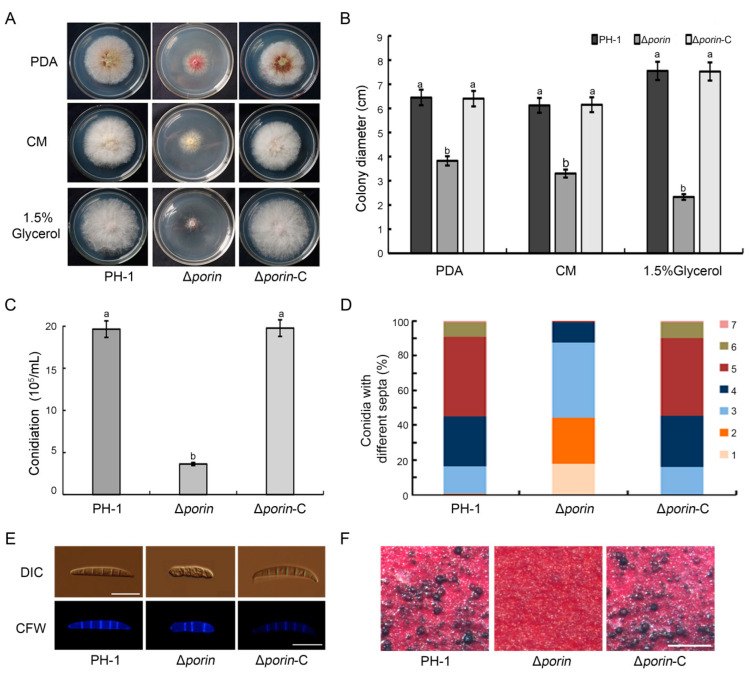
Effects of *Fgporin* on mycelial growth, conidiation, and sexual reproduction. (**A**) Colony of PH-1, ∆*porin*, and ∆*porin*-C strains on PDA, CM, and 1.5% glycerol medium at 25 °C for 3 days. (**B**) Colony diameters of PH-1, ∆*porin*, and ∆*porin*-C were examined. (**C**) Conidiation produced by PH-1, ∆*porin*, and ∆*porin*-C was counted using 4-day-old culture in CMC medium. (**D**) Conidial morphology of PH-1, ∆*porin*, and ∆*porin*-C. (**E**) Conidial septa were stained with 1 μg/mL CFW (Calcofluor white) solution and observed via fluorescence microscopy. Bar = 10 µm. (**F**) Perithecia formation in PH-1, ∆*porin*, and ∆*porin*-C on carrot agar plates. Bar = 500 µm. Values followed by the same letter are not significantly different (*p* < 0.05) using the Tukey test.

**Figure 3 jof-08-00936-f003:**
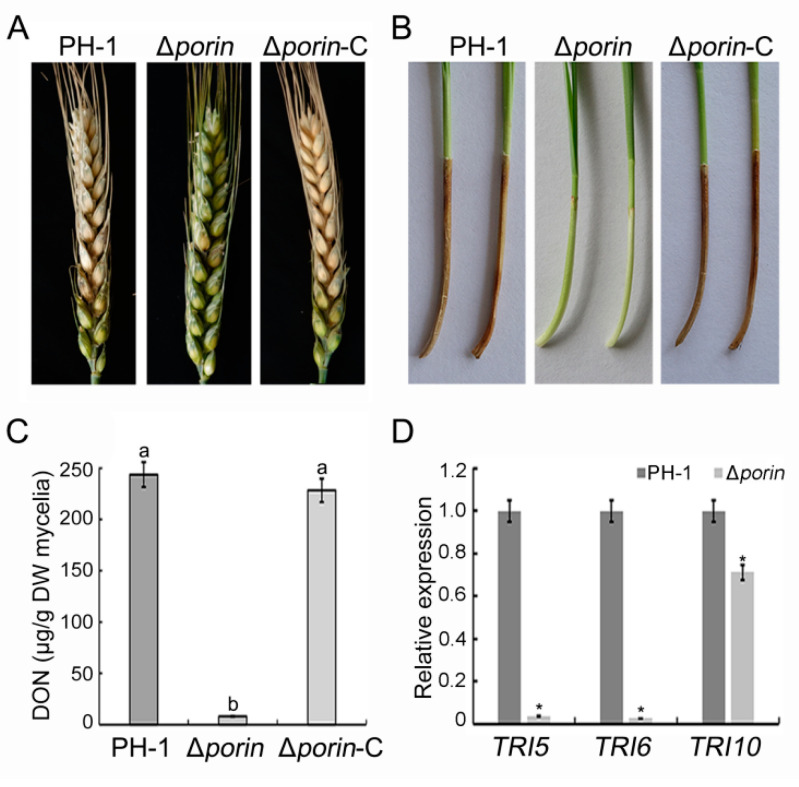
*Fgporin* regulates virulence and DON production. (**A**) Virulence of PH-1, ∆*porin*, and ∆*porin*-C strains on wheat spikelets at 14 days post-inoculation (dpi). (**B**) Virulence of PH-1, ∆*porin*, and ∆*porin*-C strains on wheat coleoptiles at 10 dpi. (**C**) PH-1, ∆*porin*, and ∆*porin*-C strains were cultured in TBI medium, and DON production was measured at 7 dpi. Values followed by the different letters are significantly different (*p* < 0.05) according to the Tukey test. (**D**) Relative expressions of *FgTRI5*, *FgTRI6,* and *FgTRI10* in PH-1 and ∆*porin* strains. The expression of *FgGAPDH* was used as an internal control. The relative expressions of *FgTRI5*, *FgTRI6,* and *FgTRI10* in PH-1 were all set at 1.0. Asterisks indicate significant differences (*p* < 0.05) using the Student’s *t*-test.

**Figure 4 jof-08-00936-f004:**
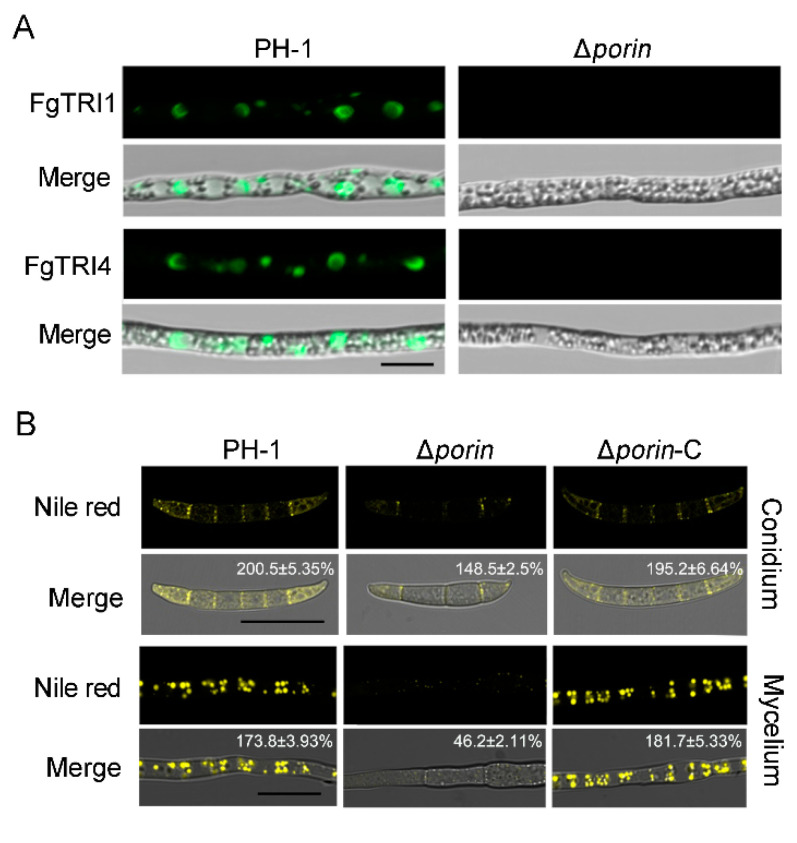
*Fgporin* regulates the formation of toxisomes and lipid droplets. (**A**) The Δ*porin* strain was defective in the formation of toxisomes. PH-1 and Δ*porin* strains tagged with plasmids carrying TRI1-GFP and TRI4-GFP were incubated in TBI medium at 25 °C for 3 days. Bar = 10 μm. (**B**) Defective LDs biogenesis in the Δ*porin* strain. PH-1 and Δ*porin* strains were incubated in TBI medium at 25 °C for 2 days. Hyphae were stained with 25 μg/mL Nile red. The numbers in the graphs represent the fluorescence intensity, and at least five fluorescent pictures were measured using ImageJ 1.53e software for each treatment. Bar = 10 μm.

**Figure 5 jof-08-00936-f005:**
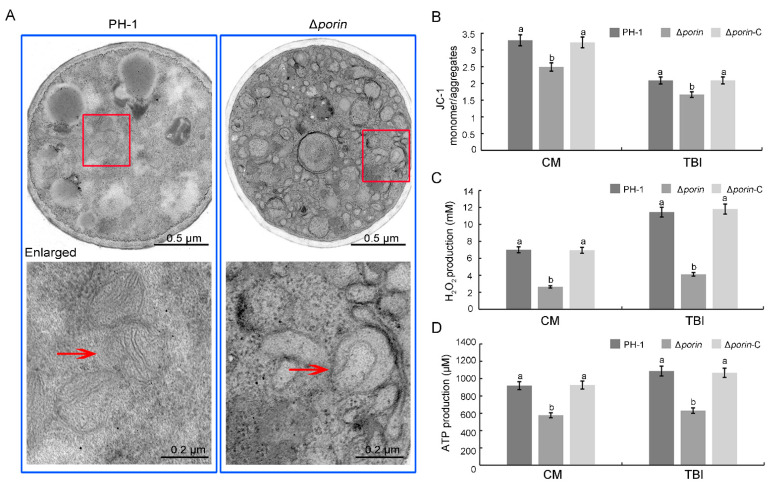
*Fgporin* causes changes in mitochondrial morphology and function. (**A**) Swollen mitochondria and fewer cristae in the Δ*porin* mutant. The ultrastructural morphology of the mitochondria in each strain was visualized via transmission electron microscopy. The cell structure indicated by the arrow was the mitochondrion. The down panel was enlarged from the red box. Bar dimensions are shown in the images. The *Fgporin* mutant reduced mitochondrial membrane potential (**B**), endogenous H_2_O_2_ production (**C**), and ATP production (**D**). Mycelia cultured in CM and in TBI were harvested for the detection of membrane potential, H_2_O_2_ and ATP, respectively. Values followed by the different letters are significantly different (*p* < 0.05) according to the Tukey test. Bar = 10 μm.

**Figure 6 jof-08-00936-f006:**
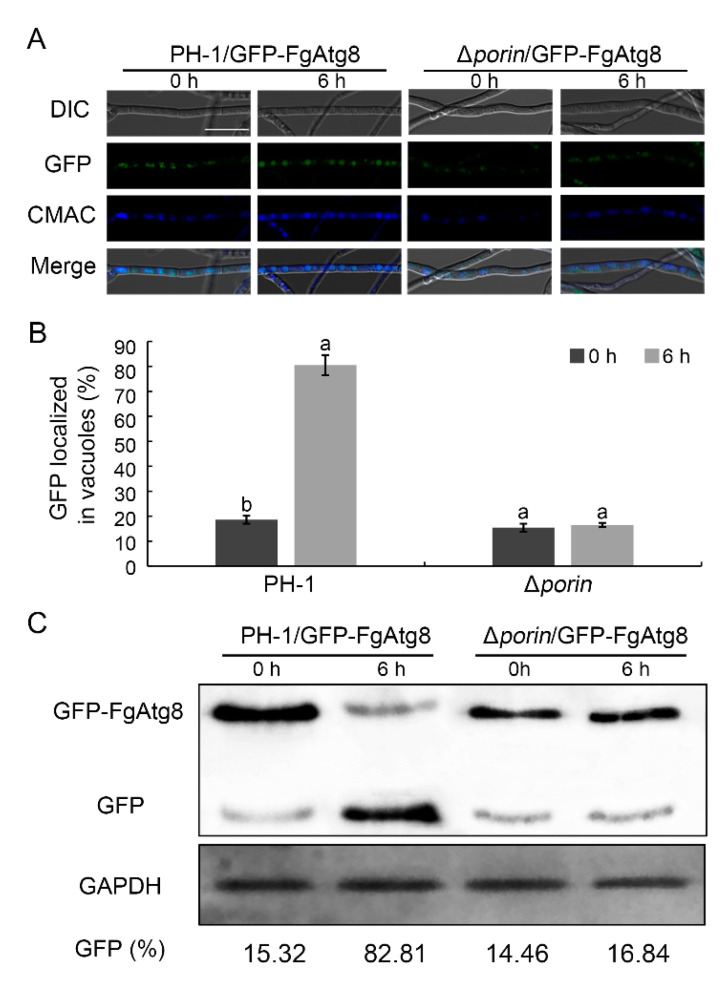
Deletion of *Fgporin* resulted in impaired autophagy. (**A**) Fluorescence assays of autophagy in the Δ*porin* mutant. PH-1 and the Δ*porin* mutant expressing a *GFP:FgAtg8* fusion gene (PH-1/GFP:FgAtg8 and Δ*porin*/GFP:FgAtg8) were subjected to MM-N medium for 6 h, stained with 10 µM CMAC (7-amino-4-chloromethylcoumarin), and then examined under a laser scanning confocal microscope. MM-N, minimal medium without nitrogen. Bar = 20 μm. (**B**) The quantitation of green fluorescence dots localized in vacuoles. Different letters indicate significant differences (*p* < 0.05) using the Student’s *t*-test. (**C**) GFP:FgAtg8 proteolysis was examined using Western blot in PH-1/GFP:FgAtg8 and Δ*porin*/GFP:FgAtg8 strains. GAPDH was used as an internal control.

**Figure 7 jof-08-00936-f007:**
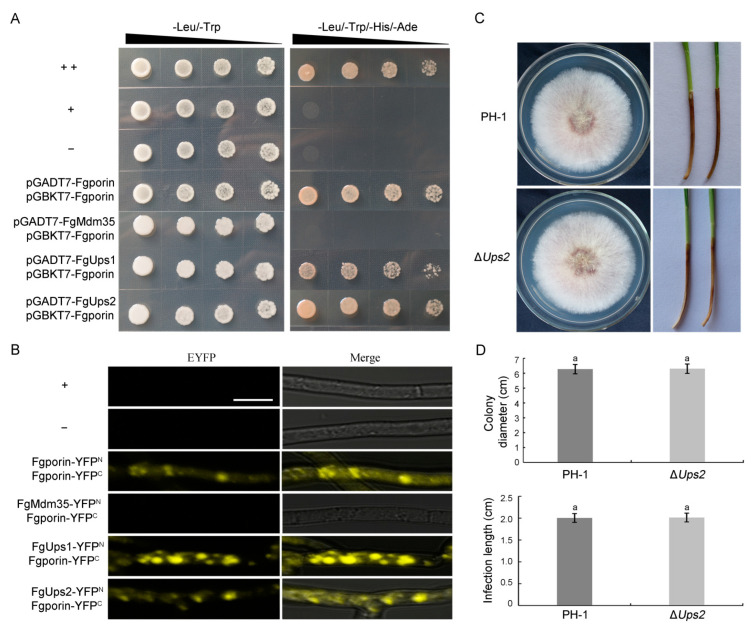
Fgporin displays self-interaction and interacts with FgUps1/2. (**A**) Yeast two-hybrid assays of Fgporin self-interaction and interactions with FgUps1/2. ‘++’: pGADT7-T + pGBKT7-53, ‘+’: pGADT7-FgMdm35/FgUps1/FgUps2/Fgporin + pGBKT7, or pGADT7 + pGBKT7-Fgporin, ‘−’: pGADT7-T + pGBKT7-Lam. (**B**) BiFC assays of Fgporin self-interaction and interactions with FgUps1/2. Hyphae were incubated in CM medium for 24 h and harvested for fluorescence examination. ‘+’: Fgporin/FgMdm35/FgUps1/FgUps2-YFP^N^ + YFP^C^, or YFP^N^ + Fgporin-YFP^C^, ‘−’: YFP^N^+YFP^C^. Bar = 10 µm. (**C**) The ∆*Ups2* mutant displayed normal colony morphology and virulence compared with PH-1. PH-1 and ∆*Ups2* were cultured on PDA medium for 3 days (left panel), and wheat coleoptiles were inoculated with conidial suspension (right panel). (**D**) The ∆*Ups2* mutant displayed the same growth and virulence compared with PH-1. Colony diameter (up panel) and infection length (down panel) were counted on PDA plates and coleoptiles, respectively. Values followed by the same letter are not significantly different (*p* < 0.05) according to the Student’s *t*-test.

**Table 1 jof-08-00936-t001:** Conidial length, disease index, and lesion size in the wild type PH-1, ∆*porin,* and ∆*porin*-C strains.

Strain	Conidial Length (μm) ^a^	Disease Index on Wheat Heads ^b^	Lesion Length on Coleoptiles (cm) ^c^
PH-1	45.41 ± 6.52 ^a^	12.4 ± 1.5 ^a^	1.82 ± 0.18 ^a^
∆*porin*	35.73 ± 5.12 ^b^	0.7 ± 0.3 ^b^	0.12 ± 0.04 ^b^
∆*porin*-C	45.32 ± 4.32 ^a^	12.6 ± 1.8 ^a^	1.88 ± 0.13 ^a^

Different superscript letters denote a significant difference (*p* < 0.05) using the Tukey test. ^a^ Conidia were harvested from 4-day-old cultures, and at least 300 conidia were counted for measurement. ^b^ The disease index on wheat heads was counted at 14 days post-inoculation (dpi). ^c^ Lesion length on wheat coleoptiles was counted at 10 dpi.

## Data Availability

Not applicable.

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
