# Peer review of "Mitochondrial Porin Is Involved in Development, Virulence, and Autophagy in Fusarium graminearum"

_jof, 2022, doi:10.3390/jof8090936_

Round 1
Reviewer 1 Report
The manuscript presented by Han et al. is a solid work on the role of mitochondrial porin in Fusarium graminearum. The manuscript presents clear and multiple defects in fungal biology, including asexual and sexual development, pathogenicity and DON production. More specifically the manuscript presents a clear defect in mitochondrial biology. The writing is clear, the work is solid and careful and conclusions are reasonable and based on the data. However, there is nothing completely surprising about the results, they are expected and it is not entirely clear how this work truly advances science.
Author Response
亲爱的审稿人博士,
我们衷心感谢您对我们的稿件进行批判性审查和评论。它们将对我们的科学研究有用。
禾谷镰刀菌是导致谷类作物(包括小麦、大麦和玉米)头枯病(FHB)的主要植物病原体。目前,控制FHB的有效策略尚不可用。因此,鉴定与真菌发育和毒力相关的发病机制有助于提供新颖有效的控制策略。在这项研究中,我们已经确定了线粒体孔蛋白同系物Fgporin,这对F. graminearum的菌丝生长,毒力和自噬很重要。Fgporin的缺失导致线粒体形态和功能的变化。这些发现将实现对Fgporin及其调节机制的全面理解作用,并帮助科学家找到潜在的真菌靶标,以有效控制FHB的策略。

Reviewer 2 Report
The manuscript by Han et al. identified the mitochondrial porin Fgporin in a pathogenic fungus Fusarium graminearum. Authors created a gene knockout of Fgporin which resulted in hyphal growth defects and reduced virulence of Fusarium graminearum. Later authors also observed defects in mitochondrial function and morphology as well as autophagy defects. Further yeast two-hybrid and BIFC assay suggest that Fgporin interacts with lipid transporter Ups1. Overall, the data support the conclusions very well and the study is well designed and executed, and I have only a few suggestions for improving the manuscript.
1. Figure 1: It would be more appropriate to create a table showing the amino acid similarities of FgPorin with other species.
2. How do authors confirm that FgPorin can function as porin? A functional assay to test a porin function or an indirect assay to test a porin function would be appropriate.
3. Is it possible to fluorescently tag FgPorin to test the localization of FgPorin?
4. Why did the authors check the expression of TRI? There is no discussion or rationale about this experiment.
5. The EM images of ∆porins look drastically different than PH-1 and it looks like there is a significant change in other organelles as well. Is it true for most cells? Is there any difference in the number of mitochondria between WT and ∆porins mutant? It would be really good to show the difference in cristae number.
6. Figure 5: The figure legends are not in the order as per the figure.
7. Figure 6A: A quantification of autophagosome would be appropriate.
8. Throughout the manuscript the species name should be italicized.
Author Response
Response to Reviewer 2 Comments
Dear Dr. Reviewer,
We thank you sincerely for critical review and comments on our manuscript. All the comments are valuable for the improvement of this manuscript. They are also useful for our future researches. Below are our responses to the comments.
1: Figure 1: It would be more appropriate to create a table showing the amino acid similarities of FgPorin with other species.
Response: Thank you very much for your comment. We have added a Table S2 showing the amino acid similarity of FgPorin with other species.
2: How do authors confirm that FgPorin can function as porin? A functional assay to test a porin function or an indirect assay to test a porin function would be appropriate.
Response: Thank you very much for your comment. In this study, we found the amino acid sequences of yeast and human porin proteins on NCBI, and used this as a target for comparison on the NCBI website through the blastp tool, and successfully identified FGSG_09933 in the Fusarium graminearum database. Fgporin is a porin protein through the amino acid alignment, secondary structure and phylogenetic tree assays.
3: Is it possible to fluorescently tag FgPorin to test the localization of FgPorin?
Response: In yeast and mammals, porins are localized in the outer mitochondrial membrane. So we haven’t done localization experiments.
4: Why did the authors check the expression of TRI? There is no discussion or rationale about this experiment.
Response: The TRI gene cluster is responsible for the biosynthesis of DON, so we investigated the expression of TRI. We added this in Discussion.
5: The EM images of ∆porin look drastically different than PH-1 and it looks like there is a significant change in other organelles as well. Is it true for most cells? Is there any difference in the number of mitochondria between WT and ∆porin mutant? It would be really good to show the difference in cristae number.
Response: Thank you very much for your suggestions. It is true that ∆porin mutant displayed a significant change than WT and all hyphal cells were consistent. For Δporin, we observed and photographed over 50 views. We feel that it was not easy and inaccurate to account the number of mitochondria in EM experiments.
6: Figure 5: The figure legends are not in the order as per the figure.
Response: We have modified the order of images in this Figure.
7: Figure 6A: A quantification of autophagosome would be appropriate.
Response: Thank you very much for your comment. We have added Figure 6B showing GFP signal in vacuoles.
8: Throughout the manuscript the species name should be italicized.
Response: We have made modifications according to your suggestions.
Round 2
Reviewer 2 Report
The revised manuscript addresses my concern and It can be accepted in its present form.